# Discovery of indole-modified aptamers for highly specific recognition of protein glycoforms

Alex M. Yoshikawa[1], Alexandra Rangel[2], Trevor Feagin[2], Elizabeth M. Chun[2], Leighton Wan[3], Anping Li[2], Leonhard Moeckl[4], Diana Wu[3], Michael Eisenstein[2,5], Sharon Pitteri [2] & H. Tom Soh [2,5,6 ✉]

Glycosylation is one of the most abundant forms of post-translational modification, and can have a profound impact on a wide range of biological processes and diseases. Unfortunately, efforts to characterize the biological function of such modifications have been greatly hampered by the lack of affinity reagents that can differentiate protein glycoforms with robust affinity and specificity. In this work, we use a fluorescence-activated cell sorting (FACS)-based approach to generate and screen aptamers with indole-modified bases, which are capable of recognizing and differentiating between specific protein glycoforms. Using this approach, we were able to select base-modified aptamers that exhibit strong selectivity for specific glycoforms of two different proteins. These aptamers can discriminate between molecules that differ only in their glycan modifications, and can also be used to label glycoproteins on the surface of cultured cells. We believe our strategy should offer a generally-applicable approach for developing useful reagents for glycobiology research.

[1] Department of Chemical Engineering, Stanford University, Stanford, CA 94305, USA. [2] Department of Radiology, Stanford University, Stanford, CA 94305, USA. [3] Department of Bioengineering, Stanford University, Stanford, CA 94305, USA. [4] Department of Chemistry, Stanford University, Stanford, CA 94305, USA. [5] Department of Electrical Engineering, Stanford University, Stanford, CA 94305, USA. [6] Chan Zuckerberg Biohub, San Francisco, CA 94158, USA. ✉email: tsoh@stanford.edu

Glycosylation, in which saccharides (i.e., glycans) are covalently linked to a target protein, is one of the most abundant and diverse forms of post-translational modification. Although once thought to be of limited importance, it has become clear that glycans have a profound effect on a wide range of biological processes. For example, glycans are known to be involved in cell adhesion, molecular trafficking and clearance, receptor activation, signal transduction, and endocytosis[1]. Alterations in glycosylation have also been implicated in many human diseases including the development and progression of cancer[1], and malignancy-specific glycans have shown promise as potential biomarkers[2]. Unfortunately, our ability to study and develop diagnostics and therapeutics based on protein glycosylation is hampered by the lack of glycan-specific affinity reagents. Antiglycan antibodies are challenging to develop due to the inherently poor immunogenicity of carbohydrates: if a glycoprotein is used as an antigen, the generated antibodies will preferentially target the protein epitopes over the glycan epitopes[3]. Furthermore, many anticarbohydrate antibodies exhibit poor selectivity, undermining their utility in the context of basic research or clinical applications[3].

Aptamers are nucleic acid-based affinity reagents that could offer an effective alternative to antibodies for the detection of protein glycosylation. Aptamers can be selected in vitro, are readily chemically synthesized, and undergo facile chemical modification[3,4]. However, it has proven extremely challenging to isolate aptamers with high specificity for particular protein glycoforms, and to date there are only two examples of aptamers targeting protein glycans in the literature[5,6]. This is likely due to several factors, including the challenges of obtaining and characterizing well-defined protein glycans for use in the selection process, and the fact that aptamers generally prefer to interact with protein epitopes over glycan epitopes[3]. Despite these challenges, a DNA aptamer was recently selected that binds to a specific glycoform of prostate-specific antigen (PSA)[5]. Another group selected DNA aptamers that were chemically modified with a boronic acid moiety, which bound to the glycoprotein fibrinogen[6]. The boronic acid moieties incorporated into these DNA aptamers were shown to form key interactions with the sugar groups present on the glycoprotein. However, boronic acid groups can be problematic due to their pH-dependent binding, as well as the fact that other molecules containing cis-diol groups can compete with the target glycan for binding[3]. To the best of our knowledge, no other aptamer chemical modifications have been tested in the context of glycan recognition.

In this work, we demonstrate that the incorporation of the aromatic heterocyclic indole moiety into base-modified DNA aptamers can enable the highly specific recognition of protein glycan epitopes. We identified indole as a promising candidate modification because tryptophan, an indole derivative, often forms key interactions within the binding pockets of anticarbohydrate antibodies[7]. Analysis of X-ray crystal structures of proteins noncovalently bound to carbohydrates has likewise revealed that tryptophan is highly enriched within carbohydrate-binding pockets relative to other amino acids[8]. Furthermore, it has been suggested that indole favorably binds to the electron-poor C-H bonds of carbohydrates, and that complementary electronic effects help drive protein-carbohydrate interactions[8]. Using an adaptation of the multiparameter particle display (MPPD) aptamer screening technique developed by our group[9,10], we were able to generate aptamers bearing indole moieties that displayed exquisite glycan specificity for multiple proteins, robustly discriminating against non-target protein variants of identical structure that differ only in terms of their glycosylation.

## Results and discussion

**Strategy for selecting an N-glycan-binding aptamer.** As a strategy for isolating glycan-specific base-modified aptamers, we built on our previously described MPPD technique[9,10], which allows for the simultaneous evaluation of aptamer affinity and specificity in a single screening experiment (Fig. 1). MPPD aptamer selection typically begins with the conversion of a solution-phase aptamer library into monoclonal aptamer particles via an emulsion PCR process. These particles are subsequently incubated with both the target and nontarget molecules that have been differentially labeled with distinct fluorescent tags. These are then subjected to fluorescence-activated cell sorting (FACS) in order to identify the subpopulation of particles that generate a strong target-specific signal, but produce minimal fluorescence associated with the nontarget label. We have modified this process by directly incorporating the uridine nucleotide 5-[(3-indolyl)propionamide-N-allyl]-2'-deoxyuridine-5'-triphosphate (5-indolyl-dUTP), which bears an indole moiety, as a substitute for thymidine during the emulsion PCR step. As described above, we incorporated this indole moiety as a base modification in the selection scheme in order to enhance the ability of the aptamer pool to form favorable contacts with glycan epitopes. We used the commercially available KOD-XL polymerase, which has previously been shown to be capable of effectively incorporating chemically-modified nucleotides[11,12]. After the FACS screening process is complete, the collected aptamer particles are subjected to a 'reverse-transcription' reaction, which converts the base-modified aptamers back to natural DNA templates. These can then be used either for a subsequent round of screening or subjected to sequencing if sufficient enrichment of target-specific aptamers has occurred.

As an initial model for testing the advantages of selecting indole-modified aptamers for protein glycan recognition, we used RNase A (RA) and RNase B (RB). These proteins are identical in amino acid sequence, and only differ at a single N-linked glycosylation site present on RB but not on RA (Fig. 2). X-ray crystallography has shown that the structures of RA and RB are identical, and that the glycan present on RB does not alter the protein's structure[13]. This is advantageous, as it allows the selection of aptamers that specifically recognize the glycan motif rather than regions of the protein that have been structurally altered as a consequence of glycosylation. Indeed, this has been suggested to be the case for the previously discovered aptamer for a glycoform of PSA[5].

RA and RB were respectively labeled with Alexa Fluor (AF) −647 and AF-488, and then incubated with the aptamer particles for 1 h. The samples were then washed and sorted via FACS. We then collected aptamer particles that produced increased signal in the AF-488 channel but no increase in the AF-647 channel, indicating binding to RB but not RA—and therefore, specific interaction with the glycan moiety present on RB. After subjecting the initial, naïve library to a pre-enrichment step consisting of five rounds of conventional SELEX against RB immobilized onto magnetic beads, we performed two rounds of MPPD selection. In the first MPPD round, we used 5 µM fluorescently-labeled RA and RB, which we then reduced to 500 nM in the second MPPD round to increase the stringency of the selection conditions. For both the pre-enriched library (after 5 rounds of SELEX) and after each MPPD round, we performed a binding assay with the aptamer particles to assess the round-to-round enrichment of RB binders in the aptamer pool (Fig. 3A). For the pre-enriched library, we used 6.8 µM AF-488-labeled RB, whereas we used 3.4 µM for the MPPD-selected pools. We observed a clear increase in the fluorescence signal between the first and last round of MPPD selection, indicating increasing RB affinity as selection progressed.

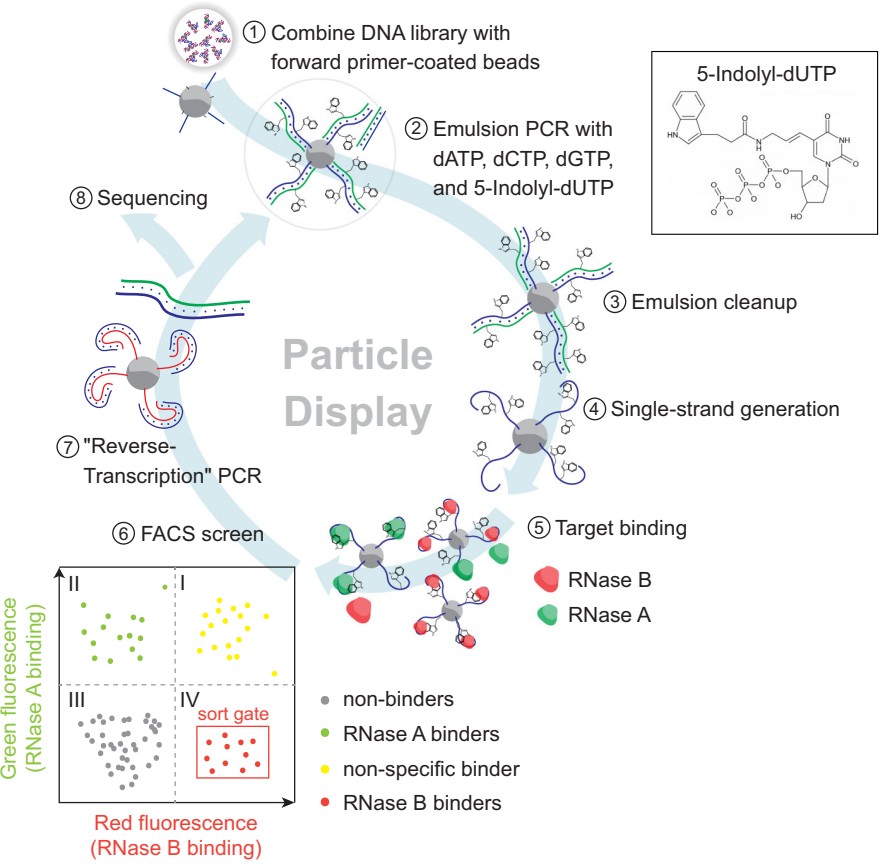

**Fig. 1 A modified multiparameter particle display (MPPD) selection scheme for generating indole-modified aptamers.** 1) DNA library molecules are hybridized to magnetic beads coated with forward primers, and 2) emulsion PCR is performed to create aptamer particles that each display many copies of a single sequence. dTTP is substituted with the modified base 5-indolyl-dUTP at this step to produce base-modified aptamers. 3) The emulsions are broken and 4) the aptamer particles are converted to single-stranded DNA via NaOH treatment. 5) The aptamer particles are incubated with RNase B (RB) and RNase A (RA), where each protein is labeled with a spectrally orthogonal fluorophore. 6) Fluorescence-activated cell sorting (FACS) separates aptamer particles that generate a strong RB-specific signal and minimal RA-specific signal (quadrant IV of the FACS plot). 7) The aptamers are subjected to a 'reverse-transcription' step to produce natural DNA, which is either used as the template for an additional round of screening or 8) characterized via high-throughput sequencing.

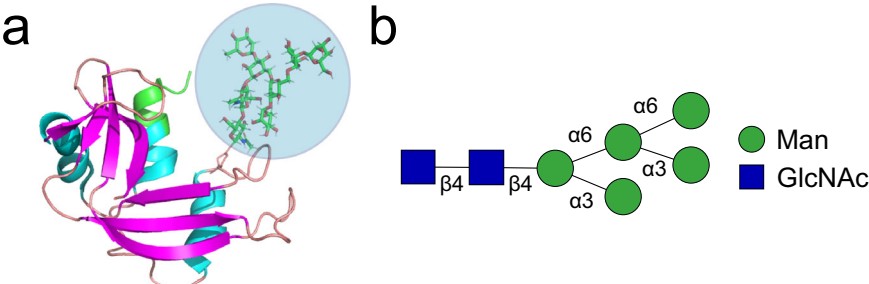

**Fig. 2 The structure of RB. A** The protein structure of RB, including a high-mannose N-linked glycan group highlighted in blue. The structure of RA is identical, save for the N-glycan. The figure was generated in PyMol using data from the PDB[13] and GlyProt[31]. **B** The N-glycan consists of mannose (Man) and N-acetylglucosamine (GlcNAc). This N-glycan is heterogeneous and can contain 1–3 additional mannose sugars via α-2 linkages[14].

We next performed two-channel binding assays to ensure that the enriched aptamer pool was specifically binding to the glycan epitope on RB rather than protein epitopes also present on RA. We incubated aptamer particles from either the naïve library (i.e., the initial library prior to pre-enrichment) or the final aptamer pool with 3.4 μM labeled RA and RB (Fig. 3B). We observed a notable shift in binding to RB between the two pools, but no increase in binding to RA relative to the initial aptamer library (quadrant IV), indicating successful enrichment for glycan-specific aptamers. Likewise, the reduction in events that display strong signal in both the fluorescence channels (quadrant I) likely reflects the elimination of aptamers that bind protein epitopes present on both forms of RNase.

**Characterization of RB-specific aptamer candidates.** Based on the clear enrichment after MPPD selection, we subjected the two aptamer pools to high-throughput sequencing (HTS). Aptamers were grouped into families using in-house Python code, where

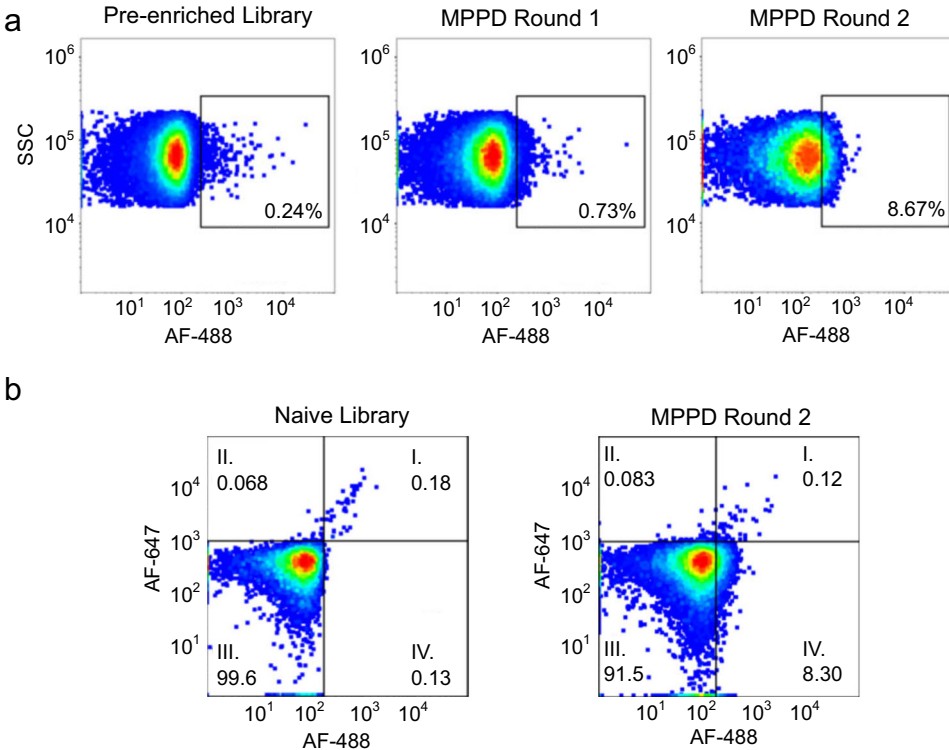

**Fig. 3 Affinity and specificity profiling of RB aptamer pools. A** Round-to-round enrichment of the aptamer pool during particle display. Aptamer particles were incubated with fluorescently-labeled RB and analyzed via flow cytometry. The pre-enriched library (after 5 rounds of SELEX) was assayed at 6.8 µM RB, while the pools obtained after MPPD rounds 1 and 2 were assayed at 3.4 µM RB. The cytometry gates are set arbitrarily to the edge of the population and provide a consistent reference point between the plots. **B** Two-color binding assays to simultaneously assess enrichment of aptamer pools for both RA and RB. Aptamer particles were incubated with 3.4 µM fluorescently-labeled RB and RA and analyzed via flow cytometry. The lefthand plot shows the naïve library (before pre-enrichment), the righthand plot shows the final pool after two rounds of MPPD. Plots were generated using FlowJo.

family members were defined as sequences having a Levenshtein distance ≤3. The MPPD round 2 aptamer pool was highly converged, with the majority of the sequences falling into two aptamer families representing 45.7% and 21.5% of the total sequencing reads (Supplementary Fig. 1). By monitoring the proportion of thymidine residues present in the aptamer sequences, we could identify enrichment of base-modified nucleotides, providing insight into their importance for glycan recognition. Although the MPPD round 2 pool did not show an overall increase in base modifications, the three most abundant families in this pool showed increased thymine content in their variable regions (30–40%). This suggests that the indole modification is an important factor enabling these aptamers to recognize the glycan motif, either directly or by forming ligand-binding structural motifs that are stabilized by hydrophobic cores.

We identified seven aptamer candidates (i-1–7) (Supplementary Table 1), representing the most abundant sequences within the three largest aptamer families, for chemical synthesis and subsequent characterization. We utilized a bead-based fluorescence assay to measure the affinity of these aptamer candidates. Briefly, we generated aptamer particles from each aptamer candidate via bead-based PCR. These were then incubated with 20 µM fluorescently labeled RA or RB and subjected to flow cytometry (Fig. 4A). Both proteins were labeled with Dylight 650 to make the direct comparison of binding between the two proteins clear, and to allow us to rule out the possibility that the aptamers were recognizing the AF-488 fluorophore conjugated to RB rather than the glycan motif. Each of the seven-candidate aptamers showed minimal RA binding, comparable with a negative control sample that contained beads conjugated to the forward primer (FP) sequence. In contrast, all seven aptamers showed a strong and significant increase in fluorescence when challenged with RB, demonstrating that our selection procedure can isolate indole-modified DNA aptamers that selectively recognize the glycan motif of RB.

Aptamer i-6 was chosen for further characterization since it displayed the strongest binding to RB of the seven aptamer candidates. We generated a full binding curve for aptamer i-6 in order to determine its $K_D$. The aptamer was incubated with fluorescently labeled RA and RB at a range of concentrations (Fig. 4B), and fitting a Langmuir isotherm to the resulting binding curve yielded a $K_D$ of 29.5 µM ± 2.7 µM. Although we observed a small increase in fluorescence at low micromolar concentrations of RA, we believe that this can be attributed to non-specific interactions as the fluorescence does not seem to further increase at even higher concentrations of RA. To demonstrate that the indole moiety was key for aptamer recognition, we synthesized aptamer i-6 with all-natural nucleotides and confirmed that no binding occurred at 100 µM RB (Supplementary Fig. 2). To demonstrate that the affinity of the aptamer was not simply due to non-specific interactions between the indole moiety and the N-glycan, we performed a binding experiment with an indole-modified DNA aptamer containing a scrambled version of the i-6 variable sequence, and demonstrated that this sequence had significantly lower binding than i-6 (Supplementary Fig. 3). To further validate aptamer binding and determine if the aptamer could function in solution (as opposed to immobilized on a particle), we performed microscale thermophoresis (MST) experiments using Cy5-labeled aptamer and unlabeled protein (Supplementary Fig. 4). The $K_D$ determined by the MST experiment for RB was virtually identical to that obtained with the bead-based binding assay (24.5 µM ± 1.2 µM). We also

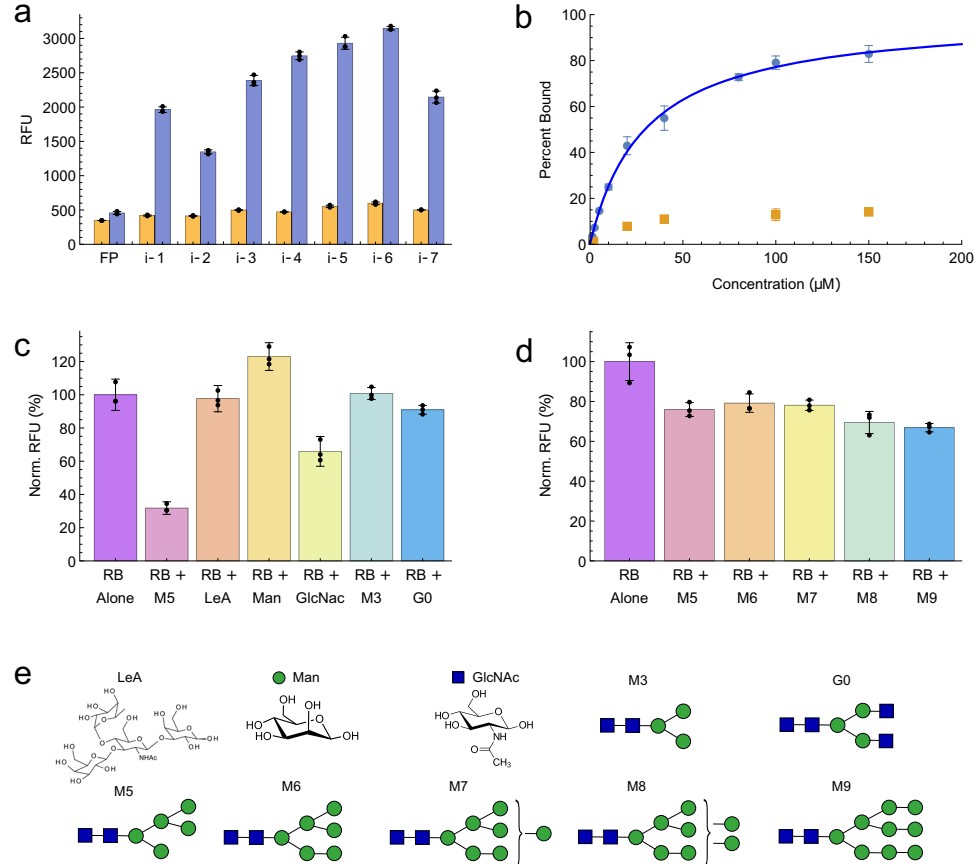

**Fig. 4 Characterization of RB aptamers. A** Aptamer candidates were converted to aptamer particles and incubated with 20 μM fluorescently labeled RA (orange) or RB (blue) prior to analysis via flow cytometry. Beads containing only the forward primer (FP) were used as a negative control. The *y*-axis shows median relative fluorescence unit (RFU) values and the error bars represent a single standard deviation of the three replicates. The raw data are overlayed on the bar plot as black dots. Source data are provided as a Source Data file. **B** Aptamer i-6 was expressed on beads and incubated with RA (orange) or RB (blue) prior to analysis via flow cytometry. The blue line represents a fitted Langmuir isotherm ($K_D = 29.5\,\mu M \pm 2.7\,\mu M$). Data are presented as median values and error bars represent the standard deviation of three replicate experiments. Source data are provided as a Source Data file. **C** Competitive binding assays for i-6 aptamer particles and 20 μM fluorescently-labeled RB alone or with 50 μM Man5 N-glycan (M5), 100 μM Lewis A (LeA) glycan, 10 mM mannose (Man), 10 mM GlcNAc, 50 μM M3, or 50 μM G0. The *y*-axis shows relative fluorescence normalized to the RB-only control. All experiments were conducted in triplicate and the median RFU values were recorded. The error bars represent a single standard deviation and the raw data points are overlayed on the bar plot as black dots. Source data are provided as a Source Data file. **D** Competitive binding assays conducted with i-6 aptamer particles and 20 μM of fluorescently labeled RB alone or with 20 μM of N-glycans containing five to nine mannose units (M5-M9). The y-axis shows relative fluorescence normalized to the RB-only control. All experiments were conducted in triplicate and the median RFU values were recorded. The error bars represent a single standard deviation and the raw data points are overlayed on the bar plot as black dots. Source data are provided as a Source Data file. **E** Structures of the molecules used in the competition assays.

examined the effect of enzymatically cleaving the glycan from RB on i-6 binding in order to control for unexpected differences between the two RNase variants. RB was deglycosylated with PNGase F, an enzyme that removes most N-linked glycans from glycoproteins, as confirmed by SDS-PAGE analysis. We then performed a single-concentration binding assay with RA, RB, and deglycosylated RB (Supplementary Fig. 5). These results confirmed that removal of the glycan essentially eliminates aptamer binding, and provide additional evidence that the aptamer is specifically interacting with the glycosylated portion of RB.

Next, we performed several competition assays to assess the glycan specificity of our newly-selected aptamer. The N-linked glycan present on RB consists primarily of mannose, and so we first assessed whether free mannose could inhibit binding between aptamer i-6 and RB due to aptamer cross-reactivity to the monosaccharide. Even at 10 mM mannose, we saw no reduction in aptamer binding to RB, suggesting that the aptamer does not interact exclusively with the carbohydrate moiety, but

rather is recognizing a larger glycan-incorporating motif (Supplementary Fig. 6). This result highlights an advantage of using the indole moiety over the boronic acid moiety, which has a tendency to broadly and nonspecifically interact with molecules containing cis-diols such as glucose, fructose, mannose, and galactose[14]. We also confirmed that the aptamer had only weak interaction with GlcNAc at concentrations up to 10 mM and we observed no reduction in binding in a competition assay using the structurally dissimilar Lewis A tetrasaccharide glycan (which does not bear the common core N-linked glycan Man₃GlcNAc₂) at concentrations of up to 100 μM, demonstrating that the aptamer does not simply bind any polysaccharide (Fig. 4C). We also observed no competition when using 50 μM of the M3 N-glycan or complex-type N-glycan G0, which suggests that the aptamer specifically recognized the high-mannose N-glycans present on RB, and not other types of N-glycans. However, when the competition assay was conducted with 50 μM of M5, an N-linked glycan that shares the same core structure as the high-mannose

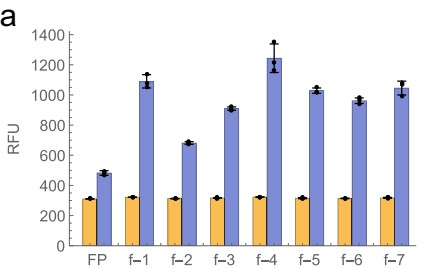
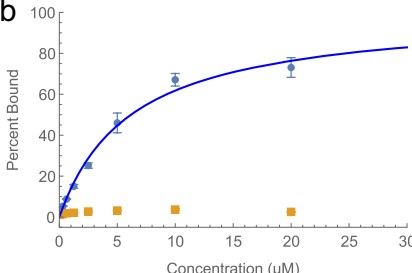

**Fig. 5 Characterization of fetuin aptamers. A** Aptamer particles displaying seven aptamer candidates were incubated with 1 μM fluorescently-labeled fetuin (blue) or asialofetuin (orange) prior to analysis via flow cytometry. Beads containing only FP were used as a negative control. The individual data points from each experiment are overlayed on the bar plot as black dots. Data are presented as median values and the error bars represent the standard deviation of three replicate experiments. Source data are provided as a Source Data file. **B** Beads coated with aptamer f-4 were incubated with either fetuin (blue) or asialofetuin (orange) prior to analysis via flow cytometry. The blue line represents the results of a fitting a Langmuir binding isotherm to the fetuin binding data ($K_D = 6.2$ μM ± 0.2 μM). Data are presented as median values and the error bars represent the standard deviation of three replicate experiments. Source data are provided as a Source Data file.

N-glycans on RB[15,16], we did observe a large decrease in aptamer binding, demonstrating that the aptamer can recognize this N-glycan in solution. To determine if the aptamer was preferentially recognizing a specific version of the N-glycans present on RB, we conducted a competition assay using equimolar concentrations (20 μM) of RB and N-glycan standards containing 5 to 9 mannose sugars (M5–9) (Fig. 4D, E). We observed a similar level of inhibition (~25% reduction) by each of these N-glycan standards, indicating that the aptamer can bind each of the high-mannose N-glycans present on RB and is not sensitive to the number of terminal mannose residues. While the aptamer could bind to the high-mannose N-glycans in solution, it displayed stronger affinity toward RB. This could be due to a number of reasons. For example, the aptamer may be interacting with protein epitopes, or it may be due to the different conformational states of solution-phase versus protein-linked N-glycans[14].

Finally, we conducted bead-based binding assays to investigate if aptamer i-6 could bind to other glycoproteins that feature various types of N-glycans (Supplementary Fig. 7). First, we incubated aptamer i-6 with 10 μM of fluorescently-labeled polyclonal goat antibodies, which are reported to contain mostly biantennary complex N-glycans[17], and did not observe any aptamer binding. Next, we incubated aptamer i-6 with 50 μM ovalbumin, which is reported to contain both hybrid and high-mannose N-glycans[18], and saw no binding. This was surprising, as we expected the aptamer to bind the high-mannose N-glycans associated with this protein, and so we conducted a mass spectrometry analysis of intact glycopeptides from ovalbumin. The analysis revealed that only 1% of the N-linked glycopeptides contained high-mannose, and that the majority of the N-glycans were complex types (Supplementary Fig. 8), which would explain the lack of binding observed. Lastly, we incubated the aptamer particles with CD2, which is known to contain the same high-mannose N-glycans (Man5-Man9) present on RB[19]. We observed a strong increase in signal, indicating binding between the aptamer and CD2 (Supplementary Fig. 7). These results, along with the competition assays, collectively indicate that the aptamer is remarkable specific towards the high-mannose N-glycans (M5-M9) present on RB, and does not bind to other complex or hybrid type N-glycans.

**Selection with fetuin demonstrates the generalizability of our selection platform.** In order to demonstrate that this is a generalizable strategy for isolating glycosylation-specific aptamers, we conducted a selection experiment for indole-modified

aptamers that specifically recognize fetuin but not asialofetuin. Fetuin is a 64 kDa glycoprotein found in the blood that is involved in multiple biological processes, such as bone remodeling and insulin resistance, and also plays a role in ischemic stroke[20]. Fetuin is heavily glycosylated, and contains sialylated N- and O-linked glycans[21], whereas asialofetuin is created by selective removal of sialic acid from fetuin[22], while retaining the same amino acid sequence, as well as the majority of the same glycans. We anticipated that creating aptamers that distinguish fetuin from asialofetuin would be more challenging than for RNase, due to the highly similar glycan motifs shared by the two proteins.

We used the same overall selection framework as for the RB-specific aptamers, with a pre-enrichment procedure comprising four rounds of SELEX against bead-immobilized fetuin and counter-SELEX against asialofetuin, followed by two rounds of MPPD. However, instead of directly incorporating the analog nucleotide containing the base-modification during the emulsion PCR step, we performed base modification after the PCR step using a click chemistry approach described in the previous work[23]. This approach enables solid-phase synthesis of the resultant aptamer using commercially available reagents, and offers more flexibility in terms of the range of base-modifications that could be incorporated into this selection scheme. Additionally, the aptamer library was changed to only include ten potential sites that could incorporate the modified nucleotide within the 50-nucleotide variable region. This was done to limit the number of indole modifications, since we observed a large number of modifications in many of the RB aptamers, which we believed could potentially interfere with some downstream applications. For the screening process, fetuin was labeled with Dylight 650 and asialofetuin was labeled with Dylight 532. We used 100 nM labeled fetuin in both rounds of MPPD, but only introduced 200 nM labeled asialofetuin in the second round (Supplementary Fig. 9). The resulting aptamer pool was then sequenced on an Illumina MiSeq.

We selected seven aptamer candidates (Supplementary Table 2) and screened them via flow cytometry with 1 μM fluorescently labeled fetuin or asialofetuin (Fig. 5A). All seven aptamers exhibited increased fetuin binding compared to a FP-only negative control, and minimal binding to asialofetuin. We selected aptamer f-4 for further characterization since it displayed the greatest increase in fluorescence when challenged with fetuin. Aptamer f-4 was one of two aptamers with the greatest number of indole moieties (4 out of 10 potential sites), providing further evidence of the benefits of the modification for glycan recognition. After incubating aptamer f-4 with fetuin and asialofetuin at a range of concentrations, we generated a binding curve and fitted

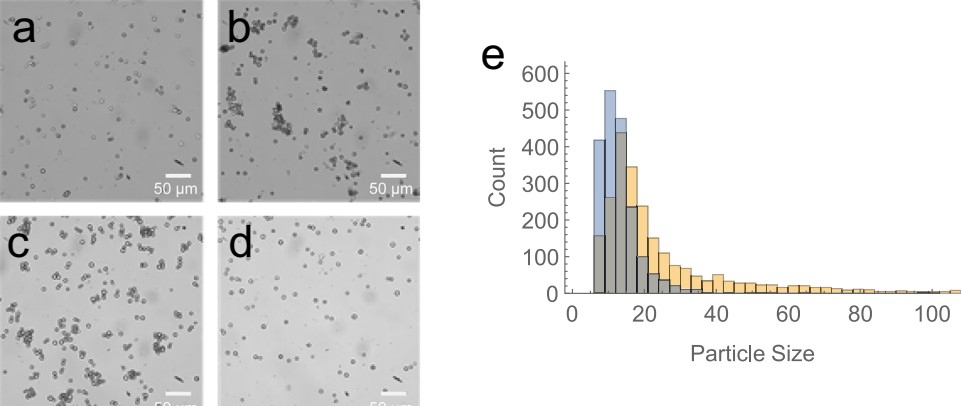

**Fig. 6 A bead-based assay to measure aptamer binding to cell-surface N-glycans.** Paraformaldehyde-fixed *Dictyostelium discoideum* cells were incubated with (**A**) streptavidin beads (negative control), (**B**) concanavalin A (ConA)-coupled beads (positive control), (**C**) aptamer i-6-coupled beads, or (**D**) aptamer i-6-coupled beads after 1 h of preblocking with ConA. Cells were imaged with a 20X objective and scale bars correspond to 50 microns. **E** Histogram of particle sizes (number of pixels) for i-6 aptamer beads incubated with fixed cells either with (blue) or without (orange) ConA blocking. Images for this analysis were taken with a 5X objective (Supplementary Fig. 10). Source data are provided as a Source Data file.

the data to a Langmuir binding isotherm (Fig. 5B). Based on these data, we measured a $K_D$ of 6.2 µM ± 0.2 µM for fetuin, with no measurable binding observed for asialofetuin, confirming the excellent specificity that can be achieved for protein glycoforms with our selection process. As a secondary validation, we performed MST using Cy5-labeled f-4 and unlabeled protein (Supplementary Fig. 10), and confirmed comparable fetuin binding ($K_D = 10.4$ µM ± 3.9 µM), with no binding to asialofetuin. To demonstrate that the indole moiety was key for aptamer recognition, we synthesized the f-4 sequence with all-natural nucleotides and observed only minimal binding to 20 µM of fetuin (Supplementary Fig. 11). Finally, we performed competition assays using 50 µM of one of the sialylated N-glycans present on fetuin (G2S2), 50 µM of the same de-sialylated N-glycan that is present on asialofetuin (G2), and 100 µM of the monosaccharide sialic acid (Supplementary Fig. 12). No reduction in signal was seen with the G2 glycan, and only minimal reductions were observed with G2S2 and sialic acid. While these results suggest that there may be interactions between the aptamer and sialylated glycans, we were unable to determine the exact nature of aptamer-fetuin recognition due to the many different types of N- and O-glycans present on fetuin and asialofetuin. It is possible that the aptamer does not recognize a single glycan, but rather forms simultaneous contacts with multiple glycans present on fetuin.

*Recognition of cell-surface glycans.* Given the exquisite specificity of aptamer i-6 toward RB versus RA, we hypothesized that this aptamer was primarily forming interactions with the glycan and its linkage to the protein via the N-glycan consensus sequence (Asn-X-Ser/Thr), rather than to the protein epitopes that are shared between RB and RA. If this hypothesis is correct, we expect some cross-reactivity with other proteins that also contain similar N-glycan structures. Therefore, we decided to conduct a proof-of-concept experiment to determine whether the aptamer could bind to the numerous N-glycans that are expressed on the surface of cells. We focused on the RB aptamers because the N-glycan on RB is well characterized, and there are protein-based reagents to probe high-mannose N-glycans, which do not exist for most glycans.

We adapted a qualitative bead-based aggregation assay[24], which is similar to assays developed for the histochemical characterization of cells based on their adhesion to beads coated with reagents such as lectins[25]. We used 10-micron magnetic

beads functionalized with streptavidin to enable facile conjugation of biotinylated i-6 aptamer. *Dictyostelium discoideum* cells were used for this experiment, as they are known to express N-glycans on their surface and have been previously used as a eukaryotic model organism for glycobiology studies[26]. The beads and paraformaldehyde-fixed cells were combined and incubated overnight, and then imaged on a confocal microscope. Previous studies have indicated that paraformaldehyde fixation does not impact the ability of lectins to bind to cells, and that these assays can produce results that are comparable to live-cell assays[25]. Unconjugated streptavidin beads and streptavidin beads functionalized with biotinylated concanavalin A (ConA) were used as negative and positive controls for the assay, respectively. We chose the lectin ConA as a positive control because it has been previously used in bead-adhesion assays[25], and is known to bind generally to high-mannose N-glycans, as well as several other monosaccharides and disaccharides[27].

Images from the negative control exhibited mostly scattered individual beads, with no interactions with the fixed cells (Fig. 6A). In contrast, the ConA-coupled beads formed dense aggregates, consistent with the expected interaction between the ConA and glycans on the cell surface (Fig. 6B). We observed similar levels of aggregation with our aptamer i-6-conjugated beads (Fig. 6C). To show that the aptamer was specifically binding to glycans on the surface of the cells, we repeated the assay with fixed cells that had been blocked with 5 mg/mL of ConA for 1 h prior to the addition of the aptamer beads (Fig. 6D). We observed no binding in these conditions, demonstrating that the aptamer is interacting primarily with glycans that are also recognized by ConA. This offers further evidence that the aptamer is specifically recognizing N-glycans, in agreement with the results from our competition assays.

In order to perform a more quantitative analysis, we utilized image processing tools in Wolfram Mathematica to detect and discriminate cells, beads, and cell-bead aggregates in our images. We imaged aptamer beads incubated with both ConA-blocked and unblocked cells using a 5X objective (Supplementary Fig. 13), which captured most of the cells and beads in a single field of view. The software then calculated the relative size of each particle group and plotted the results as a histogram (Fig. 6E). When the cells were blocked with ConA, we primarily observed single i-6-conjugated beads or cells, which showed up as particles with area less than ~25 pixels. However, when the cells were unblocked, there was a clear shift in the histogram, with many cell-bead

aggregates represented by particles with increased area. We confirmed this pattern by manually inspecting the images using ImageJ. Collectively, these results support our hypothesis that aptamer i-6 binds primarily to some combination of the N-glycan, protein-glycan linkage, and N-glycan consensus sequence, and suggest that indole-modified aptamers may be used to detect glycans expressed on the surface of cells. However, further characterization of the glycan and glycoprotein specificity of these aptamers would be essential before exploring their use in real-world applications.

In this work, we describe a workflow for the generation of indole-modified aptamers that can recognize protein glycoforms with high specificity. We have demonstrated the generalizability of the process by performing selections against two different glycoproteins, and show that the resulting aptamers do not exhibit meaningful affinity even for protein variants of identical structure that differ by just a single glycan. The aptamers developed in this study rival the binding affinity of lectins, which typically exhibit micromolar-range $K_D$ toward polysaccharides[3]. However, lectins must be identified from naturally occurring sources, whereas our aptamer selection platform allows the generation of entirely novel glycoprotein binders, with precise control over both the target and counter-target molecules. In some biological settings, lectins can achieve nanomolar affinities by assembling into homo-oligomeric structures with multiple binding sites. We believe that a similar result could be achieved with aptamers by utilizing the established practice of creating multivalent aptamers[28], thus further expanding the utility and applications of the selected aptamers. Although the specificities of the glycoprotein-binding aptamers in this study were much greater than previously reported aptamers[5,6], their affinities were notably weaker. We suspect that the higher affinities are a consequence of the lower specificities, and likely due to interactions between the aptamer and protein epitopes. However, this could also be due to the inherent properties of the various glycoproteins. We would like to note that trade-offs between affinity and specificity could be optimized by adjusting the concentration of the counter-target and gating strategy used the aptamer selection.

Aptamers that bind strongly and selectively to specific protein glycoforms could prove extremely useful as both research and diagnostic tools. This work strongly indicates that the indole moiety offers an advantageous aptamer base-modification for glycan recognition, with advantages relative to the previously characterized boronic acid functional group in terms of specificity. Although it is outside of the scope of this study, future work could investigate the mechanism by which indole-modified aptamers interact with carbohydrates, and if the mechanisms are similar to those involved in protein-carbohydrate interactions. This same screening approach could be used in the future to explore additional aptamer base modifications that could aid in the generation of glycoform-specific aptamers. In particular, it may be valuable to examine alternative hydrophobic base-modifications that have been shown to aid in aptamer-protein recognition, such as benzyl-dU and napthyl-dU[29], as well as base-modifications that are similar to other non-polar residues that have been shown to stabilize protein-carbohydrate complexes (e.g., phenylalanine, alanine, methionine, leucine, proline[30]). We selected the glycoproteins targeted in this study because they are well characterized, which was advantageous for the creation and validation of the aptamer generation pipeline. However, we see no reason based on the results obtained here that the same workflow could not be applied to a variety of other glycoprotein targets with direct clinical or diagnostic applications—for example, producing aptamers that can discriminate disease-related from healthy glycoforms. The

generation of indole-modified aptamers should offer a valuable starting point toward this goal, and it is our hope that future research will reveal additional base modifications that further expand the scope and utility of the glycobiology toolbox.

## Methods

**Materials.** N-(3-dimethylaminopropyl)-N'-ethylcarbodiimide hydrochloride (EDC; #161462) and N-hydroxysuccinimide (NHS; #130672) were purchased from Sigma-Aldrich. Methyl-PEG12-amino was ordered from Thermo Fisher Scientific (#26114). RNase A (RA) and RNase B (RB) were purchased from Sigma-Aldrich (#R6513 and #R1153). Fetuin (#F3004) and asialofetuin (#A4781) were purchased from Sigma-Aldrich. FITC-labeled polyclonal goat antibody was obtained from Abcam (#ab6840). FITC-labeled ovalbumin was obtained from Thermo Fisher Scientific (#O23020). CD2 protein was obtained from Sino Biological (#10982-H08H). The nucleotide analog 5-indolyl-AA-dUTP was ordered from TriLink Biotechnologies (#N-2065) and KOD XL polymerase was ordered from Thermo Fisher Scientific (#71-087-4). C8-alkyne-dUTP was purchased from Jena Bioscience (#CLK-T05-S). Tryptophan azide was purchased from ChemPeP (#182033). Remove-iT PNGase F and (#PO706S) and chitin magnetic beads (#E8036S) were obtained from New England Biolabs (NEB). Glyko Oligomannose 5 (Man5), Glyko Oligomannose 6 (Man6), Glyko Oligomannose 7 (Man7), Glyko Oligomannose 8 (Man8), and Glyko Oligomannose 9 (Man9) N-glycans used in the competition assay were obtained from ProZyme (#GKM-002500, −002600, −002700, −002800, and −002900). AdvanceBio Man3 (#GKR-002300), G0 (#GKC-004300), G2 (#GKC-024300), and G2S2 (#GKC-224300) were obtained from Agilent Technologies. Lewis A tetrasaccharide was obtained from Biosynth Carbosynth (#OL09843). N-acetyl-D-glucosamine (GlcNAc; #A8625), mannose (#M8574), and sialic acid (N-acetylneuraminic acid; #A0812) were purchased from Sigma-Aldrich. Fixation buffer was purchased from BioLegend (#420801). Pure-Proteome streptavidin magnetic beads were purchased from Sigma-Aldrich (#LSKMAGT02). Biotinylated concanavalin A (ConA) was purchased from Vector Laboratories (#B-1005-5) and nonbiotinylated ConA was purchased from Sigma-Aldrich (#C5275-5MG). The single-stranded DNA (ssDNA) library, primers, aptamer template sequences, and sequencing adaptor primers were ordered from Integrated DNA Technologies (IDT).

For the RB selection, the library was 90-nt long, consisting of a 50-nt variable region flanked by two 20-nt primer sites (5'-AGCAGCACAGAGGTCAGATG-N50-CCTATGCGTGCTACCGTGAA-3'). The FP sequence was 5'-AGCAGCAC AGAGGTCAGATG-3' and the RP was 5'- TTCACGGTAGCACGCATAGG-3'. The sequencing adaptors were 5'-TCGTCGGCAGCGTCAGATGTGTATAAGAG ACAGNNNNAGCAGCACAGAGGTCAGATG-3' for the FP adaptor and 5'-GTC TCGTGGGCTCGGAGATGTGTATAAGAGACAGNNNN TTCACGGTAGCAC GCATAGG-3' for the RP adaptor.

For the fetuin selection, the library was 5'-CCAGCGAGCCAGCGAC-(VNVV)₁₀-CACGCAGGACGGCACAG-3' where V was a mixed base code for A, C, or G. The FP sequence was 5'-CCAGCGAGCCAGCGAC-3' and the RP sequence was 5'-CTG TGCCGTCCTGCGTG-3'. The sequencing adaptors were 5'-TCGTCGGCAGCGTC AGATGTGTATAAGAGACAGHNNNNAGCAGCACAG AGGTCAGATG-3' for the FP adaptor and 5'-GTCTCGTGGGCTCGGAGATGTGTATAAG AGACAGN NNNCTGTGCCGTCCTGCGTG-3' for the RP adaptor.

For both libraries, 5'-amino-Spacer18-modified FP strands, 5' fluorescein-labeled FP complement strands, 5'-biotin-RP strands, and Alexa Fluor 647 RP strands were ordered from IDT and HPLC purified.

**MPPD PCR protocol.** Unless stated otherwise, the following PCR protocol was used for all experiments: 95 °C for 5 min, (94 °C for 15 sec, 54 °C for 30 sec, 72 °C for 1 min) for $X$ cycles, 72 °C for 5 min, 4 °C hold.

**Conjugation of proteins to magnetic beads.** Proteins were immobilized onto MyOne carboxylic acid magnetic beads (Thermo Fisher Scientific) according to the manufacturer's protocol for two-step coating using EDC/NHS. 300 µL of magnetic beads were washed four times with 0.1% Tween-20 in PBS, and then washed twice with 300 µL of MES (pH 6), with 10 min of incubation at room temperature (RT) on a rotator for each wash. The beads were then resuspended in 50 µL of 50 mg/mL NHS and 50 µL of 50 mg/mL EDC and incubated at RT on a rotator for 30 min. The magnetic beads were then washed twice with 300 µL cold 25 mM MES buffer (100 mM 2-(N-morpholino)ethanesulfonic acid, pH 6). For RA and RB, the beads were then resuspended in 100 µL of protein solution (2 mg/mL in PBS) and 67 µL of MES. For fetuin and asialofetuin, the beads were resuspended in 40 µL of each protein solution (10 mg/mL in PBS), 50 µL 100 mM MES, and H₂O to a final volume of 200 µL. Samples were vortexed and incubated for 1 h at RT on a rotator. The beads were then incubated with 500 µL of bead wash buffer (150 mM NaCl, 9.7 mM Tris, 9.7 uM EDTA, 0.1% Tween-20 in nuclease-free water, pH 7.5) for 15 min, washed twice with 500 µL of wash buffer, washed once with 500 µL of 0.1% Tween-20 in PBS, and then resuspended in 300 µL of 0.1% Tween-20 in PBS and stored at 4 °C.

**Pre-enrichment by SELEX for RB**. Several rounds of SELEX were conducted to enrich the library and reduce the sequence space because FACS can only screen ~$10^8$ particles in a reasonable timeframe. In the first round, natural DNA was used and no counter-selection was performed. 50 μL of RB beads were washed twice with 500 μL of selection buffer (20 mM Tris-HCl, 120 mM NaCl, 5 mM KCl, 1 mM MgCl$_2$, and 0.01% Tween-20 in nuclease-free water) and incubated for 15 min during the second wash. The beads were then resuspended in 200 μL of selection buffer with 2.5 μM naïve library for 45 min. The supernatant was removed, and 100 μL of selection buffer was added. The sample was heated at 95 °C for 5 min to elute the bound aptamers, and then the supernatant containing the aptamer pool was collected. The natural DNA library was then converted to the base-modified DNA library in a large-scale 5 mL PCR reaction (1X KOD XL buffer, 0.1 mM dATP, 0.1 mM dGTP, 0.1 mM dCTP, 0.1 mM 5-indolyl-AA-dUTP, 250 nM 5' biotinylated RP, 250 nM FP, 1 nM template DNA, and 250 U of KOD XL polymerase, brought up to volume with water). Seven cycles of PCR were performed, and the dsDNA was cleaned up using a Qiagen QIAquick PCR purification kit (#28104). 500 μL of Dynabeads MyOne Streptavidin C1 beads (Thermo Fisher Scientific, #65002) were washed twice with 500 μL of bead wash buffer and resuspended in 1 mL of bead wash buffer. The base-modified DNA was then added to the washed beads and incubated at RT on a rotator for at least 20 min. The beads were washed twice with 1 mL of wash buffer, once with 500 μL of water, and resuspended in 100 μL of water. The sample was then heated to 95 °C for 5 min, and the supernatant containing the base-modified aptamer library was collected and quantified using a Nanodrop spectrophotometer. Four additional rounds of enrichment were conducted using the base-modified aptamer library with the same process described above, except 10 μL of RB beads were used in each round and the PCR reaction was scaled down to 1 mL. In the final round, counter-selection was introduced using 5 μL of RA beads.

**Pre-enrichment by SELEX for fetuin**. Four rounds of positive and negative SELEX were performed with bead-immobilized fetuin and asialofetuin. All bead washing steps were performed using a DynaMag-2 magnetic separation stand (Thermo Fisher Scientific). 1 nmol fetuin library was diluted into 200 μL of fetuin selection buffer (100 mM NaCl, 2 mM MgCl$_2$, 5 mM KCl, 1 mM CaCl$_2$, 0.02% Tween 20, 20 mM Tris-HCl, pH 7.5), heated to 95 °C, cooled at 4 °C for 10 min, and then incubated at RT for 10 min. 4 nmol (20 μL) of fetuin-conjugated beads were washed twice with 200 μL fetuin selection buffer and then resuspended in 200 μL of folded library and incubated at RT with rotation for 1 h. Beads were washed twice with 100 μL selection buffer and eluted into 100 μL water by heating to 95 °C for 5 min twice. Recovered DNA was purified with a Qiagen MiniElute cleanup kit and eluted in 10 μL water. We performed PCR under the following reaction conditions: 10 μL 2X GoTaq PCR mix, 200 nM Fet FP, 200 nM biotinylated Fet RP, 1.5 μL recovered library, and H$_2$O up to a final reaction volume of 50 μl, using the following cycling conditions: 95 °C for 3 min, followed by X cycles of 96 °C for 15 s, 57 °C for 30 s, 72 °C for 30 s, and finally 72 °C for 2 min. To determine the correct number of cycles for amplification, a pilot PCR was run. 5 μl of the reaction was removed every 2 cycles (16 cycles total), and then run on a 10% TBE gel at 200 V for 40 min. The cycle that yielded a product of the correct length without forming undesired products was chosen for the final scaled-up PCR reaction. To generate ssDNA, biotinylated dsDNA was immobilized onto 100 μL Streptavidin C1 Dynabeads according to the manufacturer's protocol in 1X binding and washing buffer (5 mM Tris-HCl (pH 7.5), 0.5 mM EDTA, 1 M NaCl) in a 1.5 mL Eppendorf tube. Beads were incubated with 100 μL freshly prepared 0.5 M NaOH for 10 min at RT. The tube was placed on a DynaMag-2 magnetic rack for 2 min, and the supernatant was collected. Beads were washed once more with 50 μl 0.1 M NaOH. DNA was recovered from NaOH by adjusting the pH with 25 μl of 3 M NaOAc, then purified with a Qiagen MiniElute cleanup kit and eluted in 20 μl water. Rounds 2–4 incorporated negative SELEX after ssDNA generation and prior to positive SELEX. The entire ssDNA library was prepared as described above, and incubated with a 10-fold molar excess of asialofetuin beads for 1 h at RT with rotation. Beads were washed twice with 100 μL selection buffer, and the supernatant was collected and incubated with a 10-fold molar excess of fetuin beads and incubated at RT for 1 h with rotation. DNA was recovered and amplified as described above.

**FP bead conjugation protocol**. Five hundred microliter of Dynabeads MyOne Carboxylic Acid magnetic beads (Thermo Fisher Scientific) were washed five times with 500 μL of water on a magnetic rack. The beads were then resuspended in 150 μL of 0.2 mM 5' amino-Spacer18-FP, 200 mM NaCl, 1 mM imidazole chloride, and 250 mM EDC, mixed well, and sonicated prior to incubation at RT overnight on a rotator. This resulted in the covalent coupling of FP to the carboxylic acid groups on the magnetic bead surface. Next, we conjugated PEG12 to the unreacted free carboxyls on the magnetic beads through a two-step NHS/EDC reaction to reduce non-specific interaction with the target proteins. The beads were washed three times with 500 μL of 100 mM MES buffer (pH 4.7). During the last wash step, the beads were incubated for 10 min at RT on a rotator. Immediately before use, an 80 mg/mL solution of EDC and a 25 mg/mL solution of NHS were prepared in cold 100 mM MES buffer. The FP beads were then resuspended in equal volumes of NHS and EDC solutions to a final volume of 150 μL. The beads were mixed well and incubated at RT on a rotator for 30 min. The beads were washed twice with

500 μL of cold PBS. The activated beads were then resuspended in 150 μL of 20 mM amino-PEG in PBS, mixed well, and incubated for at least 30 min at RT on a rotator. The beads were then washed three times for 15 min with 500 μL of wash buffer in order to quench any amine-reactive NHS esters. Finally, the beads were resuspended in 500 μL of wash buffer and stored at 4 °C.

In order to make sure FP was successfully conjugated to the beads, 1 μL of FP beads was added to 100 μL of 100 nM fluorescein-labeled FP complement and incubated for 10 min at RT on a rotator. The beads were then washed once with 500 μL of wash buffer, resuspended in 200 μL of wash buffer, and run on a benchtop flow cytometer (BD Accuri C6 Plus).

**Emulsion PCR protocol**. The emulsion PCR process involves the creation of an oil phase and an aqueous phase. The oil phase consists of 4.5% Span-80, 0.4% Tween 80, and 0.05% Triton X-100 in mineral oil (all purchased from Sigma-Aldrich), stored at RT in the dark. The aqueous phase consists of 1X KOD XL buffer, 0.5 U of KOD XL polymerase, 0.2 mM dATP, 0.2 mM dCTP, 0.2 mM dGTP, 0.2 mM base-modified nucleotide, 10 nM FP, 1 μM RP, 2 pM dsDNA aptamer library, and ~$3 \times 10^8$ FP-coated magnetic beads (12 μL of FP-bead suspension) in a total volume of 1 mL of water. To create the water-in-oil emulsions, 7 mL of the oil phase were added to a DT-20 tube (IKA) and 1 mL of the aqueous phase was added dropwise over ~30 sec while the mixture was stirred at 600 rpm in an Ultra-Turrax device (IKA). The mixture was then stirred on the device for another 5.5 min. The emulsion was then hand pipetted into ~80 wells of a 96-well PCR plate (100 μL per well). The plate was then run on a PCR machine for 40 cycles.

**Emulsion cleanup**. After PCR the emulsions were transferred to a 50 mL Falcon tube. 125 μL of 2-butanol (Thermo Fisher Scientific) was added to each well that had contained emulsion, and the butanol was then transferred to the same 50 mL tube. The tube was vortexed for 30 sec rigorously and then centrifuged at 3000 x *g* for 6 min. A magnet was used to retain the pellet of aptamer particles while removing the supernatant. 1.2 mL of breaking buffer (100 mM NaCl, 1% Triton X-100, 10 mM Tris-HCl, pH 7.5, and 1 mM EDTA) was added to the particles, and the mixture was transferred to a new 1.5 mL tube. The 1.5 mL tube was vortexed and centrifuged at 21,000 x *g* for 1 min. Using a magnetic rack, the supernatant was removed with a 1 mL micropipette. Another 1 mL of breaking buffer was added to the particles and removed as described above for multiple cycles until absolutely no white film was visible on the top of the supernatant. 400 μL of breaking buffer was then added, and the sample was transferred to a new 1.5 mL tube. Once again 1 mL of breaking buffer was added, vortexed, and centrifuged at 21,000 x *g* for 1 min. The supernatant was removed, and the aptamer particles were resuspended in 1 mL of wash buffer.

**ssDNA generation**. The aptamer particles were resuspended in 800 μL of 100 mM NaOH and incubated for 10 min at RT on a rotator. The aptamer particles were washed twice with 800 μL of 100 mM NaOH and then three times with 1 mL of wash buffer, briefly mixing between each wash. Finally, the aptamer particles were resuspended in 200 μL of wash buffer.

**Aptamer particle quality control**. To ensure the successful synthesis and monoclonality of the aptamer particles, 1 μL of the aptamer particle solution, 98 μL of wash buffer, and 1 μL of 100 nM Alex Fluor 647-modified RP was added to a 1.5 mL tube. The tube was incubated for 10 min at RT on a rotator and then washed once with 500 μL of wash buffer. The sample was then resuspended in 200 μL of wash buffer and run on a flow cytometer (BD Accuri C6 Plus, Software Version 1.0.23.1). Monoclonality was assessed as previously described by our group[9].

**Protein labeling**. RA, RB, and CD2 were labeled using Dylight protein labeling kits (Thermo Fisher Scientific). After the labeling reaction was complete, the samples were concentrated using a 3 K Amicon Ultra-0.5 mL Centrifugal Filter (EMD Millipore) and then dialyzed in PBS using a 3.5 K Slide-A-Lyzer MINI Dialysis Device (Thermo Fisher Scientific) overnight. The samples were then dialyzed in selection buffer for 4 h and concentrated again using 3 K Amicon filters. Finally, the absorbance of the DNA and the fluorophore were determined using a Nanodrop spectrophotometer, and the degree of protein labeling was determined using the formulas provided by the kit manufacturer.

Fetuin and asialofetuin were labeled by standard amine-NHS ester coupling. 0.3 mg fetuin and 0.3 mg of asialofetuin were separately resuspended in 0.05 M sodium borate buffer and incubated with 25 μg DyLight 650 NHS ester or Alexa Fluor 532 NHS ester (Thermo Fisher Scientific) for 1 h at RT. Labeled proteins were purified by dialysis (Slide-A-Lyzer MINI 10 K; Thermo Fisher Scientific) overnight, and the degree of protein labeling was determined using a NanoDrop spectrophotometer.

**MPPD screening protocol for RB-specific aptamers**. During each round of MPPD, ~$10^8$ aptamer particles (the entire aptamer particle stock except for the material used for quality control) was incubated with equimolar concentrations of labeled RA and RB (5 μM for the first round, 500 nM for the second round). Prior

to incubation, the aptamer particles were resuspended in 500 µL of selection buffer and incubated for 15 min at RT on a rotator. After incubation, the beads were sonicated and resuspended with 250 µL of fluorescently labeled protein in a selection buffer. The aptamer particles were incubated for 1 h in the dark at RT on a rotator, washed once with 500 µL of cold selection buffer, resuspended in 3 mL of cold selection buffer, and put on ice. The aptamer particles were sorted on a BD Aria II FACS instrument (using BD FACSDiva software, version: 8.0.2) using a 75-micron nozzle, and gating on the forward and side scatter was used to ensure that only singlet beads were analyzed. Around 0.1–0.3% of the singlet aptamer population was collected that displayed the greatest shift in RB binding (FITC channel) without any increase in RA binding (APC channel). The aptamer particles were kept at 4 °C in the FACS sample chamber. After sorting, the collected aptamer particles were transformed back to natural dsDNA as previously described[9].

Finally, each aptamer pool was assessed for the enrichment of RA and RB binders. 1 µL of the aptamer particle solution was added to fluorescently tagged RA and/or RB in selection buffer in a total reaction volume of 25 µL. The samples were incubated in the dark for at least 45 min at RT on a rotator. Each sample was individually washed with 500 µL of cold selection buffer, resuspended in 200 µL of cold selection buffer, and run on a benchtop cytometer (BD Accuri C6 Plus) at a slow flow-rate. The singlet population was identified by gating in the side-scatter and forward-scatter plots, and the fluorescent signal of the population was then analyzed to assess binding.

**MPPD screening protocol for fetuin-specific aptamers**. FP-modified beads were synthesized as described above. Aptamer particles were generated by emulsion PCR using an oil phase composed of 4.5% Span 80, 0.45% Tween 80, and 0.05% Triton X-100 in mineral oil. The aqueous phase consisted of 1x KOD XL DNA polymerase buffer, 50 U KOD XL DNA polymerase, 0.2 mM dATP, 0.2 mM dGTP, 0.2 mM dCTP, 0.2 mM C8 alkyne dUTP, 10 nM Fet FP, 1 µM Fet RP, 1 pM template DNA, and 12 µl FP beads. 1 mL of the aqueous phase was added to 7 mL of oil phase and emulsified at 620 rpm for 5 min in an IKA DT-20 tube using the IKA Ultra-Turrax device. The emulsion was pipetted into 100 µL reactions in a 96-well plate and amplified using the protocol: 95 °C for 3 min, 39 cycles of 96 °C for 15 s, 57 °C for 30 s, and 74 °C for 1 min, followed by 72 °C for 5 min. Emulsion cleanup and aptamer particle quality control cleanup were performed as described for RA/RB. Aptamer particles recovered from the emulsion PCR were washed twice with 200 µL 1X PBS and resuspended in a solution containing 1X PBS, 5 µL of a pre-prepared mixture of 0.1 M CuSO4/0.2 M tris(3-hydroxypropyltriazolylmethyl) amine (THPTA), and 10 mM tryptophan azide, and H2O to a final volume of 200 µL. A 50 mM solution of sodium ascorbate was freshly prepared in H2O, and 20 µL was added to the reaction mixture for a final concentration of 5 mM. The solution was degassed using N2 for 5 min, then reacted for 40 min with rotation at room temperature. ssDNA was generated by resuspending the particles in 200 µL 0.1 M NaOH and incubating for 10 min at RT. Beads were washed three times and resuspended in a fetuin selection buffer.

One round of MPPD was performed with Dylight 650-labeled fetuin and Alexa Fluor 532-labeled asialofetuin. Prior to sorting, a flow cytometry binding assay was performed with multiple concentrations of fetuin (10, 50, 100, 200 nM) to determine which resulted in sufficient binding to the target. A second binding assay was then performed using the concentration of fetuin determined in the first assay in the presence of multiple concentrations of asialofetuin (10, 50, 100, 200 nM). For the assay, 1 µL of the aptamer particle solution was added to a solution containing the labeled protein targets at the specified final concentrations in a total reaction volume of 50 µL of 1X fetuin selection buffer and incubated in the dark for 1 h at RT on a rotator. Samples were then washed with 200 µL and resuspended in cold 1X fetuin selection buffer for analysis. The optimal concentration of asialofetuin was determined as the highest concentration at which >1% binding to Dylight 650-fetuin remained. For sorting, aptamer particles were folded in 1 mL fetuin selection buffer and then incubated with 100 nM Dylight 650-labeled fetuin and 200 nM Alexa Fluor 532-labeled asialofetuin on a rotator in the dark for 1 h at room temperature. The beads were washed twice and resuspended in 1 mL cold 1X selection buffer and then analyzed using a BD FACS Aria III. The sort gate was set to collect 0.3% of aptamer particles that showed high specificity for fetuin by identifying those particles with the greatest shift in the APC channel (fetuin) and no overlap in the PE channel (asialofetuin). After sorting, the collected aptamer particles were resuspended in 20 µL PBS, and the aptamers were amplified by PCR using the conditions described above.

**High-throughput sequencing protocol**. For each aptamer pool sequenced, adaptor primers were first added. 10 ng of dsDNA was subjected to eight cycles of PCR. A 2x GoTaq Master Mix was used (Promega, M7132) with 1 µM of each primer (100 µL reaction volume). The sequencing primers were added by using a Nextera XT kit (Illumina) and following the provided instructions. Samples were quantified using a Qubit fluorometer and sent to the Stanford Functional Genomics Facility (SFGF) for sequencing on an Illumina MiSeq.

**Family analysis of sequencing data**. All analyses were conducted in Python. First, the number of replicates were counted for each sequence, and sequences with $N$ or

more replicates were selected ($N = 3$ for RB, 2 for fetuin). Using the levenshtein_distance function from the Levenshtein Python library, the edit distance between all the sequences was calculated and stored. The sequences were then clustered into families of edit distance of 3 or less, and sorted based on the abundance of family members.

**Synthesis of aptamer particles**. Aptamers were coated onto beads by preparing a 100 µL PCR reaction consisting of 10 µL 10X KOD XL buffer, 1 µL dNTP mix of 10 mM of each nucleotide (including the modified nucleotide analog), 0.5 µL of 10 µM FP, 2.5 µL of 100 µM RP, 5 µL of 10 nM aptamer template, 8 µL of FP beads, 71 µL of water, and 2 µL of KOD XL polymerase. 30 PCR cycles were conducted. The beads were washed and converted to ssDNA as described above for the ePCR protocol, and resuspended in 50 µL of selection buffer prior to storage at 4 °C.

**Bead-based affinity measurements**. A 10 µL binding reaction was set up with 2 µL of aptamer particle solution, the required volume of the fluorescently-labeled protein stock in selection buffer, and then brought up to 10 µL in selection buffer. For the RB competition assays, competitor molecules were added as a concentrated stock solution in selection buffer to the reaction mixture immediately before adding the fluorescently-labeled protein stock. The samples were incubated on a rotator at RT for 45 min and then put on ice. Two microliter of the reaction was added to 200 µL of cold selection buffer and put on a magnet for 1 min. The supernatant was discarded, and the particles were washed with 500 µL of cold selection buffer and then resuspended in 200 µL of cold selection buffer. The sample was gently mixed via pipette and then immediately run on a flow cytometer (BD Accuri C6 Plus). Each replicate within an experiment refers to an independent incubation using the same protein and aptamer particle stocks.

**Microscale thermophoresis (MST) affinity measurements**. A Monolith Pico Red (Nanotemper Technologies) instrument was installed with MO.Control v2.1 software was used to conduct the MST experiments. Analysis was done using MO.Affinity Analysis v3.0.4. For i-6 characterization, 1 nM 5' Cy5-labeled aptamer was incubated with varying concentrations of unlabeled RA or RB (20 µL reaction volume). The reaction was incubated for 1 h while rotating at RT, and then each sample was loaded into Monolith NT.115 Capillaries (Nanotemper Technologies, #M0-K022). A binary binding affinity measurement assay was conducted, following onscreen instructions. Data points identified as aggregating or having inconsistent initial fluorescent values were classified as outliers and removed from the analysis. The f-4 aptamer was similarly assayed against fetuin and asialofetuin, except with 5 nM Cy5-labeled aptamer. Each replicate consisted of an individual incubation with the same protein stock and labeled aptamer. Dissociation constants for each replicate experiment were calculated assuming a binary binding model with Nanotemper's MO.Affinity Analysis.

**PNGase F deglycosylation of RNase B**. Five micoliter of Remove-iT PNGase F (NEB), 3 µL of selection buffer, and 12 µL of a 40 µM RB stock in selection buffer were added to a 1.5 mL tube and incubated at 37 °C for 15 h while shaking at 350 RPM. Magnetic chitin beads (NEB) were then used to remove the PNGase F, which contains a chitin-binding domain (CBD). 50 µL of chitin beads were washed twice with 200 µL of selection buffer, and then combined with the deglycosylation reaction. The mixture was incubated for 30 min on a rotator at RT, and then the supernatant was collected. The deglycosylated RB was stored at 4 °C. Each replicate consisted of an individual incubation using the same stock of aptamer particles and proteins.

**LC-MS/MS analysis of ovalbumin**. Ten micrograms of ovalbumin protein were used to perform LC-MS/MS analysis. Disulfide bonds were reduced by adding 1 µL of 200 mM Tris(2-carboxyethyl) phosphine (TCEP) (Sigma-Aldrich) in 50 µL of 50 mM ammonium bicarbonate (Sigma-Aldrich) and incubated at 65 °C for 1.5 h. Free thiol groups were alkylated with 1.5 µL of 200 mM iodoacetamide (Acros Organics) with 45 min incubation at room temperature in the dark. Reduced and alkylated protein was digested with 1 µg of sequencing grade modified trypsin enzyme (Thermo Fisher Scientific) for 18 h at 37 °C. The resulting tryptic peptides were dried using a speed vacuum (LabConco) and reconstituted with 30 µL of 0.1% formic acid (Fisher Scientific) in HPLC MS grade water (Fisher Scientific) for LC/MS-MS analysis.

A Dionex Ultimate Rapid Separation Liquid Chromatography system (Thermo Fisher Scientific) was used to load 3 µL of the reconstituted tryptic peptides onto a C18 trap column (Thermo Fisher Scientific) with a flow rate set at 5 µL/min for 10 min. Tryptic peptides were separated by reverse-phase chromatography on a 25-cm-long C18 analytical column (New Objective) packed in-house with BEH C18, 130 Å, 1.7-µm particle size (Waters) encapsulated in a column heater (MSWIL) at 60 °C. Peptides were eluted by changing the mixture of mobile phase A (0.1% formic acid in water) and mobile phase B (0.1% formic acid in acetonitrile). The gradient program consisted of holding mobile phase B at 2% for the first 10 min, slowly ramping up to 35% over the next 50 min, followed by an increase to 85% over 5 min with a five-minute hold. The analytical column was re-equilibrated for 10 min prior to the next sample injection. The flow rate throughout the gradient was set to 0.3 µL/min and each sample was analyzed in triplicate. Eluted peptides

were analyzed using an Orbitrap Eclipse Tribrid mass spectrometer (Thermo Fisher Scientific). The cycle time was set at top speed for 3 s, with an MS1 mass scan range of 375-2,000 m/z and a resolution of 120,000. The most abundant precursor ions were fragmented with higher energy collision-induced dissociation (HCD) and with collisional energy set to 38%. Dynamic exclusion was enabled for 15 s, and the normalized AGC target set to 250%. MS2 fragments were detected in the Orbitrap with a mass resolution of 30,000, with injection time set to auto.

The resulting raw data files were searched using Byonic software 4.0.12 (Protein Metrics) against the reference chicken (*Gallus gallus*) proteome (2021; 2,296 entries) and the 309-member mammalian glycan library provided in the Byonic software. Parameters included trypsin digestion with a maximum of two missed cleavages and precursor mass tolerance of 10 ppm. Fixed cysteine carbamidomethylation and variable methionine oxidation and asparagine deamination were also specified.

Identified ovalbumin glycopeptides (see the Supplementary Data file) from triplicate injections were combined and filtered to remove identifications with log probability <2, score <150, and >1 glycan identified. Glycopeptides were further classified by putative glycan type, including complex undecorated (no fucose or sialic acid), complex fucosylated, complex sialylated, complex fucosylated + sialylated, high mannose, or other (HexNAc2).

**Bead-based cell adhesion assays and analysis.** 3-4 mL of *Dictyostelium discoideum* cells (dictybase, strain AX2, ID #DBS0238585) (~$10^6$ cells per mL) grown in HL5 axenic media (10 g peptone, 10 g glucose, 0.35 g yeast extract, 0.35 g $Na_2HPO_3.7\ H_2O$, 0.35 g KHPO$_4$, 0.05 g dihydrostreptomycin-sulfate in 1 liter of water. All components were purchased from Sigma-Aldrich) at RT were pelleted via centrifugation (1580 RCF for 3 min) and then resuspended in 3 mL of paraformaldehyde fixation buffer (BioLegend) and incubated for 20 min at RT in the dark. The fixed cells were then washed twice with 1 mL of PBS prior to resuspension with 200 μL of PBS. The fixed cells were stored at 4 °C for no more than 2 weeks.

Positive control ConA beads were created by washing 20 μL of PureProteome streptavidin magnetic beads three times with 500 μL of selection buffer. The beads were resuspended in 40 μL of selection buffer, mixed with 10 μL of a 10 mg/mL stock solution of biotinylated ConA, and incubated for 30 min at RT on a rotator. The ConA-conjugated beads were then washed three times with 500 μL of PBS and resuspended in 40 μL of PBS.

The i-6 aptamer beads were created by first PCR amplifying the aptamer with biotinylated FP. A PCR reaction consisting of 5 μL of biotin-FP (100 μM), 5 μL RP (100 μM), 7.5 μL of aptamer template (100 nM), 50 μL of 10X KOD XL buffer, 10 μL dNTP mix containing 10 mM indolyl dUTP, 10 μL of KOD XL polymerase, and 412.5 μL of nuclease-free water was run for eight PCR cycles. The PCR reaction was cleaned up using a QIAquick PCR purification kit and incubated with 10 μL of PureProteome beads for 3 h in 40 μL of PBS at RT on a rotator. The aptamer beads were then washed three times with 500 μL of PBS and resuspended in 20 μL of PBS.

For the binding experiments, 5 μL of fixed cells, 1 μL of beads, and 5 μL of selection buffer were pipette mixed in a 1.5 mL tube and then incubated at RT overnight. For the experiments with ConA-blocked cells, 5 μL of the fixed cells were first incubated with 5 μL of 10 mg/mL ConA for 1 h. Once the incubation was complete, 5 μL of the cell-bead mixtures were added to a glass microscope slide and imaged via confocal microscope. No coverslip was added, since we observed that the coverslip or excess pipetting could disrupt the bead-cell aggregates.

Images were collected with a 5X objective and captured using Micro-Manager (version 1.4). Images were analyzed using Wolfram Mathematica (Version 12.0.0.0). Briefly, the image processing function ComponentMeasurements was used to determine the pixel areas of beads, cells, and bead-cell aggregates and then plotted as a histogram. The Mathematica code used to analyze and plot the images are provided in the Supplementary Information.

**Reporting summary**. Further information on research design is available in the Nature Research Reporting Summary linked to this article.

## Data availability
The data underlying Figs. 4–6, as well as Supplementary Figs. 2, 3, 4, 5, 6, 7, 10, 11, and 12 are contained in the Source Data file. The mass spectrometry data for the ovalbumin glycoprotein is contained in the Mass Spectrometry Data file. Any additional data from this study is available from the authors upon reasonable request. Source data are provided with this paper.

## Code availability
The Mathematica code used for image processing is contained within the supplementary information file.

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

## Acknowledgements

We would like to thank Dr. Chan-Ho Park for providing the cultured cells used in this study and we would like to thank Abel Bermudez for conducting the mass spectrometry glycopeptide analysis of ovalbumin. This work was supported by the Chan-Zuckerberg Biohub, Bill and Melinda Gates Foundation and the National Institutes of Health (NIH, OT2OD025342). A.M.Y. is supported by a Stanford Bio-X Graduate Fellowship.

## Author contributions

A.M.Y., T.F., and H.T.S. designed the initial RNase B experiments. A.M.Y., E.M.C, and A.L conducted the RNase B aptamer selections. A.R. and D.W. conducted the fetuin aptamer selection. L.W. and A.M.Y analyzed the high-throughput sequencing data. A.M.Y, L.M., and S.P. designed the aptamer-glycan binding characterization experiments. A.M.Y. performed the aptamer-glycan characterization experiments. S.P conducted the mass spectroscopy experiments. A.M.Y., H.T., and M.E. wrote and edited the manuscript.

## Competing interests

The authors declare no competing interests.
