## [Peer Review File · Nature Communications]

Reviewers' Comments:

Reviewer #1:

Remarks to the Author:

The manuscript by Yoshikawa et al describes the discovery and characterization of indole-modified aptamers capable of specific recognition of protein glycoforms. Using particles display, an in vitro screening approach previously developed by the Soh group, the authors identified two lead aptamers that exhibit strong selectivity for specific glycoforms of two different proteins. Importantly, the lead aptamers identified in this work contain indole modifications, which the authors argue enable the highly specific recognition of protein glycan epitopes. Glycan recognition has historically been challenging for aptamers (and in general). Thus, this work does represent an important advance in the aptamer field and I expect will be of broad interest to the readers of Nature Communications. However, I feel that the central claim of this work – that the indole moiety “enable the highly specific recognition of protein glycan epitopes” – requires additional support (see comment 5). I recommend this manuscript for publication after the authors address this concern and my other comments below:

1. In the SI, the authors state that several rounds of traditional SELEX were carried out prior to MPPD due to the limited throughput of FACS. This pre-selection should be discussed in the main text. There is also some confusion regarding round numbering. For example, is round 0 in the main text for the RB aptamer actually the 6th round of total selection? Some clarification is needed.
2. Statements like “Based on the clear enrichment after two rounds of selection,...” and “Although the round 2 pool...” are misleading because 7 rounds of selection were actually carried out in this example. In all cases, the pre-selection rounds should be included when discussing the number of rounds used.
3. When the authors discuss the “naïve” library (e.g. Figure 3B), is this the initial library or the library following pre-enrichment?
4. The sequence of i-6 is nearly 50% indole-modified dUTP. Is it possible that having a high level of indole moieties results in a modest affinity for N-glycans regardless of aptamer sequence? It will be important to demonstrate that scrambled control of i-6, containing the same number of indole modifications, is unable to bind RB.
5. Aptamers isolated against fetuin have very few indole modifications (Table S2). F-1 has none at all, yet still binds fetuin with comparable affinity to f-4. It is important that the authors show that the fetuin aptamers require the indole moiety to bind fetuin. If not, these results argue against their primary assertion that the indole moiety is essential for selective recognition of protein glycoforms.
6. As far as I can tell, all characterization experiments described in this manuscript were carried out with aptamer particles (i.e. immobilized aptamers). Do the free, non-immobilized aptamers bind their targets? This is an important point because a requirement that the aptamers be bound to beads potentially limits the utility of this approach.
7. Some discussion of aptamer affinity is warranted. For example, are the K_d values reported herein adequate for potential downstream applications, such as detection of Fetuin in the blood? How does the affinity of these indole-modified aptamers compare to previously reported aptamers targeting protein glycans?
8. Chemical structures of the competitor glycans in Figure 5D would be helpful.

Reviewer #2:

Remarks to the Author:

In the manuscript “Discovery of indole-modified aptamers for highly specific recognition of protein glycoforms”, Yoshikawa et al. present a particle-display-based workflow to generate and screen for aptamers containing indole-modified bases for recognizing and binding with glycosylated proteins. By using the workflow, the authors produced several aptamer candidates, which can distinguish glycosylated RNase B (RB) from non-glycosylated RNase A (RA). They showed that the interaction between RB and aptamer candidates was not significantly affected by free mannose, Man5, and Lewis A glycans. The authors further showed the workflow's potential to distinguish differently glycosylated fetuins. Finally, the authors suggest the RB-recognizing aptamer can cross-react with

other glycoproteins by showing the aggregation of aptamer-coupled beads on paraformaldehyde-fixed *Dictyostelium discoideum* cell surface.

While the topic is of great interest to protein glycosylation studies, the developed workflow lacks a comprehensive characterization of the binding epitopes of the screened aptamers. It remains unclear which part(s) of the target proteins were recognized and bound by the aptamers, which limits its applications. The successful generation of aptamers discriminating sialylated and asialylated fetuins is encouraging, but the characteristics of the binding epitopes are again missing. The bead aggregation experiment, unfortunately, in my opinion, leaves more questions than it answers. A good demonstration of the developed aptamer's potential applications in glycobiology will significantly strengthen the manuscript.

Specific comments/questions:

1. The authors should comprehensively characterize the binding epitopes of the generated aptamers. For instance, free Man5 reduced the interaction of RB and the RB-specific aptamer by ~20% (Figure 4D). Whether the aptamer can bind to other high-mannose-type glycans? Does the aptamer recognize a specific glycan type attached on RB? If not, can the authors specifically define which glycans the aptamer can bind and with which affinities?
2. The K_d of the protein-glycosylation-recognizing aptamer is at the low- μ M level, which is not better than existing lectins. The authors may want to provide evidence to demonstrate the advantages of aptamers.
3. The bead aggregation experiment (Figure 6) does not disclose what exactly the aptamer beads bound on the cell surface. The authors should provide more convincing evidence to show the aptamer can recognize protein glycoforms at a global level.
4. In the aptamer screening workflow, would it be possible to use all glycoproteins enriched from cell lysate (or even live cells) with and without a specific glycosidase treatment as the binding targets? A highly specific glycosidase may help screen aptamers recognizing a specific glycoform.

Revision response:

Reviewer 1: The reviewer felt that this work would be of broad interest to the readers of Nature Communications, but also suggested a number of important points for clarification and additional control experiments prior to publication. We greatly appreciate the reviewer's thoughtful comments and recommendations, and have addressed them as detailed below:

1. **“In the SI, the authors state that several rounds of traditional SELEX were carried out prior to MPPD due to the limited throughput of FACS. This pre-selection should be discussed in the main text. There is also some confusion regarding round numbering. For example, is round 0 in the main text for the RB aptamer actually the 6th round of total selection? Some clarification is needed.”**

We completely agree, and have added a description of the pre-selection to the main text and now make an explicit distinction between the rounds of pre-selection and the rounds of particle display. Figure 3 has been updated accordingly.

2. **“Statements like “Based on the clear enrichment after two rounds of selection,...” and “Although the round 2 pool...” are misleading because 7 rounds of selection were actually carried out in this example. In all cases, the pre-selection rounds should be included when discussing the number of rounds used.”**

We agree that the references to selection rounds were previously unclear, and believe that the changes made in response to comment #1 should address any ambiguities.

3. **“When the authors discuss the “naïve” library (e.g. Figure 3B), is this the initial library or the library following pre-enrichment?”**

This refers to the initial library, prior to pre-enrichment. We have added a sentence clarifying this in the manuscript.

4. **“The sequence of i-6 is nearly 50% indole-modified dUTP. Is it possible that having a high level of indole moieties results in a modest affinity for N-glycans regardless of aptamer sequence? It will be important to demonstrate that scrambled control of i-6, containing the same number of indole modifications, is unable to bind RB.”**

We agree that this is an important consideration, and have run a scrambled-sequence control that has a randomized variable region containing the same distribution of each nucleotide. The scrambled sequence exhibited low levels of binding relative to the forward-primer negative control, indicating that the modification indeed has some baseline ability to interact with N-glycans. Nevertheless, the scrambled control showed significantly weaker RB binding than i-6, confirming that the aptamer sequence and structure is critically involved in glycan recognition. These results have been added to the SI (Figure S3, shown below), with a sentence describing the experiment in the main text.

5. **“Aptamers isolated against fetuin have very few indole modifications (Table S2). F-1 has none at all, yet still binds fetuin with comparable affinity to f-4. It is important that the authors show that the fetuin aptamers require the indole moiety to bind fetuin. If not, these results argue against their primary assertion that the indole moiety is essential for selective recognition of protein glycoforms.”**

To address the reviewer’s suggestion, we ran a control experiment with a natural DNA version of the f-4 aptamer that does not contain indole modifications. This natural DNA sequence showed similar performance to the forward-primer negative control, confirming that the indole is necessary for fetuin recognition. We have added these data to the SI (Figure S8, shown below) and added a sentence regarding this control experiment to the main text.

It should be noted that the affinity of the fetuin aptamers does seem to correlate with the presence of the modification. We believe that the reviewer is referring to f-2 rather than f-1, which contains two indole-modified groups; f-2 lacks an indole moiety and had the lowest binding of all aptamers tested in that assay, while f-4 and f-7, which contained the most modifications, had the strongest binding. We would also like to point out that the random region used for the fetuin selection was designed to include fewer indole moieties (VNVV)₁₀, such that the presence of the four modification present in aptamer f-4 is greater than would be expected due to chance. We realize that this was not discussed clearly in the main text, and so we have added a description of the library used for the fetuin selection to the revised text.

6. “As far as I can tell, all characterization experiments described in this manuscript were carried out with aptamer particles (i.e. immobilized aptamers). Do the free, non-immobilized aptamers bind their targets? This is an important point because a requirement that the aptamers be bound to beads potentially limits the utility of this approach.”

We thank the reviewer for this thoughtful comment. We have utilized microscale thermophoresis (MST) to interrogate the binding between Cy5 labeled aptamer and unlabeled glycoprotein target in solution. First, we examined the RB aptamer i-6, and found consistent K_D measurements for the bead-based ($\sim 29 \mu\text{M}$) and MST ($\sim 25 \mu\text{M}$) assays. We also conducted the assay for RA, and observed weak binding for i-6 ($K_D \sim 150 \mu\text{M}$). We were surprised to see evidence of binding to RA, as we observed no signal in bead-based assays at RA concentrations of up to $150 \mu\text{M}$. However, we believe that the discrepancy is understandable considering the many differences between the two binding assays and the fact that MST assay is sensitive to fluorophore’s environment.

We also conducted MST with aptamer f-4 for both fetuin and asialofetuin ($n = 3$). The MST results confirmed binding of the aptamer to fetuin, with a similar K_D of $\sim 10 \mu\text{M}$ (versus $\sim 6 \mu\text{M}$ for the bead-based assay) and no measurable binding to asialofetuin.

We believe these MST experiments demonstrate that the aptamers can function in solution, and provide a valuable confirmation of binding of both the RB and fetuin aptamers. These experiments have been added to the SI section (Figures S4 and S9).

- 7. “Some discussion of aptamer affinity is warranted. For example, are the K_D values reported herein adequate for potential downstream applications, such as detection of Fetuin in the blood? How does the affinity of these indole-modified aptamers compare to previously reported aptamers targeting protein glycans?”**

We have added discussion regarding previous aptamers towards protein glycans and the implication of their K_{DS} to the manuscript. Briefly, we measured K_{DS} similar to those observed for lectins (which are multimers) as well as aptamers and antibodies selected for glycans. These affinities are several orders of magnitude weaker than aptamers selected for glycoproteins in two previously-published reports cited in the manuscript (Díaz-Fernández *et al.* 2019, Li *et al.* 2008). However, the specificities of those aptamers are much lower, and we believe that the higher affinities are likely due to the aptamer forming more interactions with the protein epitopes. It is also possible that there are simply inherent differences between the glycoprotein targets themselves. We would like to note that if higher affinity is desired and less specificity is necessary, this method could be tuned by adjusting the concentrations and gating strategy for the counter-target. We selected the glycoproteins targeted in this study because they are well characterized, which was advantageous for the creation and validation of the aptamer generation pipeline, but believe this workflow can be applied to create aptamers towards a wide variety of other glycoprotein targets.

- 8. “Chemical structures of the competitor glycans in Figure 5D would be helpful.”**

The structures of the competitor molecules are now shown in Figure 4E. Please note that this figure has been modified to include five additional competitor molecules.

Reviewer 2:

The reviewer was generally enthusiastic about the topic and our aptamer discovery methodology. However, s/he felt that the initial manuscript needed more comprehensive characterization of the binding epitopes of the screened aptamers. We thank the reviewer for the support and have addressed these concerns in the revision.

- 1. “The authors should comprehensively characterize the binding epitopes of the generated aptamers. For instance, free Man5 reduced the interaction of RB and the RB-specific aptamer by ~20% (Figure 4D). Whether the aptamer can bind to other high-mannose-type glycans? Does the aptamer recognize a specific glycan type attached on RB? If not, can the authors specifically define which glycans the aptamer can bind and with which affinities?”**

We agree that it is unclear exactly what the aptamers are binding to, and that additional epitope characterization is necessary. To answer this question, we extensively expanded the competition assays conducted in the manuscript. First, we obtained four additional high-mannose N-glycan standards that are present on RB and repeated the binding assay to determine if the aptamer was sensitive to these terminal branching mannose residues. We found that each of these N-glycan standards demonstrated similar inhibition of binding, indicating that the aptamer is not sensitive to the terminal mannose residues and perhaps recognizes the GlcNac core of the N-glycan. Second, since competition from the Man5 N-glycan was minimal at 20 μ M, we increased the concentration to 50 μ M to provide a more compelling case that aptamer binding is inhibited due to binding to the free form of the Man5 N-glycan. We observed a substantial decrease in signal, confirming that the aptamer is also binding to the glycan, albeit with a slightly lower affinity. Finally, although we had determined that the aptamer could not be inhibited by very high concentrations (up to 10 mM) of the free mannose monosaccharide, we never investigated whether the N-acetyl glucosamine (GlcNac) monosaccharide, which is present in the core of the N-glycan, could inhibit binding. We therefore conducted a competition assay with 10 mM GlcNac, and observed slight inhibition of aptamer binding to RB, indicating modest affinity for the GlcNac monosaccharide. As a whole, these experiments suggest that the aptamer is recognizing part of the conserved N-glycan core, including GlcNac residues.

Finally, we would like to note that we wanted to create a truly comprehensive “map” of specificity for each glycan structure using the strategy described above. Unfortunately, we quickly learned that this is not feasible because well-defined glycans are simply not available for many of the structures of our interest, due to the challenges associated with their synthesis. Given this, we believe the above additional experiments provide sufficient insights into the RB aptamer’s binding epitopes, and have added these competition assay results to the main text (Figure 4C–E, shown below) as well as a discussion of the results.

2. “The K_d of the protein-glycosylation-recognizing aptamer is at the low- μM level, which is not better than existing lectins. The authors may want to provide evidence to demonstrate the advantages of aptamers.”

We believe that the primary advantage our aptamer selection system provides over lectins is that they can be generated through an *in vitro* selection process to generate entirely novel reagents, whereas lectins must be identified from molecules that already exist in nature. We would also like to note that lectins are typically expressed as multimers, which greatly enhances their K_D through avidity, and typically have monovalent affinities in the very high micromolar to millimolar range. We have added some discussion regarding the benefits of aptamers over lectins to the Conclusion section of the manuscript.

3. “The bead aggregation experiment (Figure 6) does not disclose what exactly the aptamer beads bound on the cell surface. The authors should provide more convincing evidence to show the aptamer can recognize protein glycoforms at a global level.”

We believe that the additional competition binding assay experiments provide solid evidence that the aptamer can recognize high-mannose N-glycans, which we believe

provides sufficient evidence that the aptamer can recognize N-glycans in a global context.

- 4. “In the aptamer screening workflow, would it be possible to use all glycoproteins enriched from cell lysate (or even live cells) with and without a specific glycosidase treatment as the binding targets? A highly specific glycosidase may help screen aptamers recognizing a specific glycoform.”**

We agree that the reviewer’s suggestion would be an excellent and worthwhile idea for a future experiment, and although this would fall outside the scope of the current manuscript, we strongly believe that the workflow developed in this project could serve as the foundation for such an effort.

Reviewers' Comments:

Reviewer #1:

Remarks to the Author:

The authors have adequately addressed my prior comments and concerns. I now recommend this work for publication in Nature Communications.

Reviewer #2:

Remarks to the Author:

1. The additional competition binding assays provide a better characterization of the aptamer binding. However, further validation is still needed for unambiguous identification of the aptamer's binding area, which, in my opinion, is crucial for this work. The authors suggested that the RB-specific aptamer binds to the N-glycan core structure. If it is the case, this aptamer is not able to differentiate different N-glycans and should recognize other types of N-glycans as well, such as hybrid or complex types. The authors can/should further validate their suggestion, for instance, by performing binding assays using other glycoproteins that bear different N-glycans (such as fetuin) with or without specific glycosidases treatment.

2. Following the first point, if the RB-specific aptamer recognizes the N-glycan core structure and does not distinguish different glycan structures (such as Man5-9), its application is limited to the separation of proteins with or without N-glycosylation but not the proteins bearing a specific glycan. The authors need to provide more data to fully reflect their claim that "we use a fluorescence-activated cell sorting (FACS)-based approach to generate and screen aptamers with indole-modified bases, which are capable of recognizing and differentiating between specific protein glycoforms.". I suggest the authors also perform an in-depth characterization of the sialic acid-specific aptamer.

3. I do not entirely agree with the statement that "We believe that the additional competition binding assay experiments provide solid evidence that the aptamer can recognize high-mannose N-glycans, which we believe provides sufficient evidence that the aptamer can recognize N-glycans in a global context.". The presented data only show the interactions between the aptamer and high-mannose glycans but not anything else. It remains unclear whether the aptamer binds to other types of N-glycan on the cell surface. In fact, I do not find evidence demonstrating the aptamer-cell interaction was N-glycan-specific and was not mediated via any other molecules on the cell surface. Figure 6 is not sufficient to conclude.

Reviewer #1: “The authors have adequately addressed my prior comments and concerns. I now recommend this work for publication in Nature Communications.”

We thank the reviewer for recommending our work for publication.

Reviewer #2:

1. **“The additional competition binding assays provide a better characterization of the aptamer binding. However, further validation is still needed for unambiguous identification of the aptamer’s binding area, which, in my opinion, is crucial for this work. The authors suggested that the RB-specific aptamer binds to the N-glycan core structure. If it is the case, this aptamer is not able to differentiate different N-glycans and should recognize other types of N-glycans as well, such as hybrid or complex types. The authors can/should further validate their suggestion, for instance, by performing binding assays using other glycoproteins that bear different N-glycans (such as fetuin) with or without specific glycosidases treatment.”**

We agree with the reviewer that additional insight into the binding of the RB aptamer towards different types of N-glycans would strengthen the manuscript. First, we conducted additional competition experiments using both an N-glycan containing only 3 mannose sugars (M3), as well as the complex-type native N-glycan G0. We showed that even at 50 μ M concentrations, there was no significant competitive aptamer binding by either of these glycans in assays with 20 μ M labeled RB. This shows that the aptamer is specifically recognizing high-mannose N-glycans, which typically contain 5–9 mannose residues attached to the chitobiose core. These data have been added to Figure 4C.

Furthermore, we investigated the binding of the i-6 aptamer to three different fluorescently-labeled glycoproteins: 10 μ M polyclonal antibody, 50 μ M ovalbumin, and 25 μ M CD2. We saw no binding to the polyclonal antibody, which should contain a mixture of different types of N-glycans, or to ovalbumin, which is known to contain both high-mannose and hybrid structures. Since we expected the aptamer to be able to bind high-mannose N-glycans, we performed mass spectrometry analysis to investigate intact glycopeptides from ovalbumin, and determined that only a very small percentage (~1%) of the glycopeptides contained high-mannose N-glycans, which would explain the lack of binding. Finally, we observed binding of the aptamer to CD2, which is known to contain the same high-mannose N-glycans (Man5-Man9) as RB (doi: [10.1126/science.7544493](https://doi.org/10.1126/science.7544493)). These results are in agreement with the competition experiments and demonstrate that the aptamer is specifically recognizing high-mannose N-glycans, and not cross-reacting with other types of N-glycans. These results have been added to the SI section as figures S7 and S8. Taken as a whole, these results show that aptamer binds to high-mannose N-glycans, but not other types of N-glycans. We believe these competition assays and glycoprotein binding assays constitute thorough characterization of aptamer i-6, and should address the reviewer's query.

2. **“Following the first point, if the RB-specific aptamer recognizes the N-glycan core structure and does not distinguish different glycan structures (such as Man5-9), its application is limited to the separation of proteins with or without N-glycosylation but not the proteins bearing a specific glycan. The authors need to provide more data to fully reflect their claim that “we use a fluorescence-activated cell sorting (FACS)-based approach to generate and screen aptamers with indole-modified bases, which are capable of recognizing and differentiating between specific protein glycoforms.” I suggest the authors also perform an in-depth characterization of the sialic acid-specific aptamer.”**

The competition and glycoprotein assays shown in the previous comment demonstrate that the aptamer is specifically recognizing high-mannose N-glycans, and does not non-specifically bind other types of N-glycans. We have updated the manuscript accordingly.

Additionally, we have performed competition assays with the fetuin aptamer f-4 using several types of N-glycans as well as the monosaccharide sialic acid. In an assay with 20 μM labeled fetuin, we saw no competitive reduction in binding in the presence of 50 μM asialylated N-glycan G2. We saw minimal competition from 50 μM of the sialylated version of the N-glycan or 100 μM sialic acid. These results demonstrate that there is some weak interaction between the aptamer and the sialylated N-glycan, but that the binding to fetuin is much stronger. These results have been added to the SI section as figure S12.

We believe these competition assays provide some useful characterization of the fetuin aptamer. However, we would like to note that doing an in-depth characterization of the aptamer is simply not feasible—there are many types of O-glycans and N-glycans present on fetuin, and it is not realistic for us to test them all. Additionally, due to the large number of sialylated glycans, it is possible that the aptamer is interacting with a combination of sialylated glycans. RB was chosen as the first target because it has a well-established and relatively simple glycosylation pattern that would enable us to characterize aptamer-glycan interaction. The purpose of the fetuin selection is to demonstrate the generalizability of the method, and show that we can develop aptamers that can differentiate between more complex differences in a protein’s glycosylation. We therefore believe that determining the exact nature of the fetuin-aptamer interaction is outside the scope of the present work.

3. **“I do not entirely agree with the statement that “We believe that the additional competition binding assay experiments provide solid evidence that the aptamer can recognize high-mannose N-glycans, which we believe provides sufficient evidence that the aptamer can recognize N-glycans in a global context.”. The presented data only show the interactions between the aptamer and high-mannose glycans but not anything else. It remains unclear whether the aptamer binds to other types of N-glycan on the cell surface. In fact, I do not find evidence demonstrating the aptamer-cell interaction was N-glycan-specific and was not mediated via any other molecules on the cell surface. Figure 6 is not sufficient to conclude.”**

We believe the glycoprotein binding assays shown above in comment #1 provide strong evidence that the aptamer is able to bind high-mannose N-glycans on glycoproteins other than RB, and does not bind other complex N-glycans. The additional competition assays further support these observations, and taken as a whole, we believe we have provided sufficient evidence that the aptamer can bind other high-mannose N-glycans.

We agree that it is important to control for the possibility that other molecules, such as proteins or other glycans on the surface of the cell, are not mediating aptamer binding. However, we believe that the results demonstrating that aptamer binding can be eliminated by blocking the cells with Concanavalin A (ConA) offer very compelling evidence that the aptamer-cell interaction is not mediated by non-specific interactions or other molecules on the cell surface. ConA is a lectin that is known to bind high-mannose N-glycans, and if aptamer binding was mediated by other molecules, we would not expect such a high degree of blocking. Although ConA also binds to some other glycans containing mannose, we believe it is unlikely the aptamer is recognizing those glycans based on the results of our competition and glycoprotein experiments.

Reviewers' Comments:

Reviewer #2:

Remarks to the Author:

I appreciate the revision. The added binding assays nicely demonstrate the specificity of the aptamer and further support the cell-surface binding experiments. I have no further questions.